# Screening Refractory Dye Degradation by Different Advanced Oxidation Processes

**DOI:** 10.3390/molecules30030712

**Published:** 2025-02-05

**Authors:** Imane Ouagued, Marc Cretin, Eddy Petit, Geoffroy Lesage, Abderrahmane Djafer, Abdallah Ouagued, Stella Lacour

**Affiliations:** 1Water–Environment Laboratory, Hassiba Benbouali University, Chlef 02000, Algeria; imane.ouagued@yahoo.fr (I.O.); ab-dj@hotmail.fr (A.D.); abouagued@yahoo.fr (A.O.); 2European Institute of Membranes, IEM UMR-5635, University of Montpellier, ENSCM, CNRS, Place Eugène Bataillon, 34095 Montpellier cedex 5, France; marc.cretin@umontpellier.fr (M.C.); eddy.petit@umontpellier.fr (E.P.); geoffroy.lesage@umontpellier.fr (G.L.)

**Keywords:** electrochemical advanced oxidation processes, electro-Fenton, anodic oxidation, coupling process, ecotoxicity, mineralization, rhodamine B degradation pathways

## Abstract

This study investigated Rhodamine B (RhB) degradation by electro-Fenton (EF), anodic oxidation (AO), and their combination (EF/AO), using a carbon felt cathode coupled to a sub-stoichiometric titanium dioxide Magnéli phase (Ti_4_O_7_) anode or a platinized titanium (Ti/Pt) anode. The results indicated that operational parameters influenced the kinetics of electrochemical reactions. An increase in current density from 10 to 50 mA cm^−2^ significantly enhanced the RhB degradation rate; 30 mA cm^−2^ was the optimal current density, balancing both energy efficiency and degradation performance. Moreover, higher RhB concentrations required longer treatment. The Microtox^®^ bioluminescence inhibition test revealed a significant toxicity decrease of the dye solution during electrochemical degradation, which was highest with EF/AO. Similarly, total organic carbon removal was highest with EF/AO (90% at pH 3), suggesting more efficient mineralization of RhB and its by-products than with EF or AO. Energy consumption remained relatively stable with all oxidation processes throughout the 480 min electrolysis period. High-resolution mass spectrometry elucidated RhB degradation pathways, highlighting chain oxidation reactions leading to the formation of intermediates and mineralization to CO_2_ and H_2_O. This study underscores the potential of EF, AO, and EF/AO as effective methods for RhB mineralization to develop sustainable and environmentally friendly wastewater treatment strategies.

## 1. Introduction

The significant water consumption by the textile industry and the complex chemical composition of its effluents contribute substantially to environmental pollution [1,2]. These effluents frequently contain recalcitrant and toxic synthetic dyes that require advanced treatment technologies for their effective degradation and detoxification. The persistence of toxic by-products poses considerable risks to both human health and aquatic ecosystems [3]. Rhodamine B (RhB), a xanthene dye extensively employed by the textile industry and with many other applications (biological stain and color additive) [4,5], was used as a model pollutant in this study (Table 1). RhB is a toxic and carcinogenic compound and is prohibited in food and cosmetics, but it is still detected in various everyday consumer products (sauces, spices, make-up products) [6,7].

Following the Globally Harmonized System of Classification and Labelling of Chemicals (GHS), RhB is classified with the GHS05 pictogram, indicating corrosiveness. It carries the health hazard statements H302 (acute oral toxicity) and H318 (serious eye damage) and the environmental hazard statement H412 (harmful to aquatic life with long-lasting effects) [8]. Its toxicity has been well described: its half maximal effective concentration (EC50) for inhibiting the growth of algae is 14 mg L^−1^, and its median lethal dose (LC50) for crustaceans and zebrafish is 24 mg L^−1^. For aquatic tracing experiments, the maximum allowable RhB concentration is 140 µg L^−1^ to prevent severe environmental effects [9]. Water pollution by RhB is associated with various environmental and health risks. It changes water color even at low concentrations (~1.0 mg L^−1^), which makes it unsuitable for domestic use [10]. RhB is also hazardous to *Cyprinodon variegatus* at a lethal concentration of 83.9 mg L^−1^ [11].

Advanced Oxidation Processes (AOPs) represent a promising class of wastewater treatment technologies. AOPs generate highly reactive oxygen species (ROS), such as hydroxyl radicals (HO^•^), to mediate the oxidative degradation of organic micropollutants [12]. AOPs include photocatalysis [13], UV photolysis [14], Fenton oxidation [15], UV ozonation [16] and photoelectrocatalytic oxidation [17]. Electrochemical advanced oxidation processes (EAOPs), such as electro-Fenton (EF) and anodic oxidation (AO), offer significant advantages [18,19]. The main advantage of electrochemical techniques over chemical techniques, and particularly EF over Fenton oxidation, is that with EF, hydrogen peroxide (H_2_O_2_) (required to initiate the Fenton reaction) is no longer needed because it is generated by reduction of dissolved oxygen in a two-electron mechanism on carbon-based electrodes.

AO typically employs non-active anodes, such as boron-doped diamond, lead dioxide, and sub-stochiometric titanium dioxide of the Magnéli phase (Ti_4_O_7_), because of their high overpotential for dioxygen formation. Non-active anodes provide a stable surface for electron transfer without participating chemically in the reactions, thus facilitating the generation of ROS, such as hydroxyl and hydroperoxyl radicals (HO^•^, HO_2_^•^), superoxide anion (O_2_^•−^), sulfate ion (SO_4_^•−^) or singlet oxygen (^1^O_2_) [20,21]. The Ti_4_O_7_ anode is a non-active anode that facilitates the generation of substantial quantities of physisorbed HO^•^ radicals (Equation (1)), thus promoting the efficient degradation and mineralization of organic contaminants [22] while minimizing perchlorate species formation, compared with the boron-doped diamond anode [23]. However, AO alone may not be sufficient for the removal of very recalcitrant pollutants and/or intermediate products.M + H_2_O → M (HO^•^) + H^+^ + e^−^(1)

EF allows the continuous electro-generation of H_2_O_2_ via the two-electron reduction of molecular dioxygen, which is really efficient on carbon-based materials (Equation (2)). This is one of the reasons that led to the use of commercial carbon felt as the cathode in this study. The oxidative potential of H_2_O_2_ is strongly amplified in the presence of Fe^2^⁺ within an acidic medium, resulting in the production of highly reactive hydroxyl radicals (HO^•^) through the Fenton reaction (Equation (3)).O_2_ + 2H^+^ + 2e^−^ → H_2_O_2_(2)Fe^2+^ + H_2_O_2_ → Fe^3+^ + HO^•^ + HO^−^(3)

Fenton’s reagent can be electrochemically generated in the solution, with simultaneous H_2_O_2_ production by cathodic reduction of the dissolved molecular dioxygen and regeneration of the ferrous ions consumed by the Fenton reaction [24], as described in Equations (2) and (4).Fe^3+^ + e^−^ → Fe^2+^(4)

For effective EF implementation, it is essential to optimize H_2_O_2_ electro-generation [25]. The operating parameters are also of crucial importance (e.g., acidic medium, choice of electrode material, applied current density, oxygen source, Fe^2+^ catalyst concentration).

Various active anodes can be coupled to the carbon-based cathode in EF. Ti/Pt anodes exhibit high electrocatalytic activity. Dimensional stable anodes, such as IrO_2_/Ti and RuO_2_/Ti, provide good efficiency and stability, although they are expensive due to the iridium and ruthenium components [26]. Active anodes have low overpotential for O_2_ formation and are not able to produce reactive radical species (ROS) through solvent oxidation.

More interestingly, coupling EF and AO (EF/AO) may capitalize ROS generation by using a non-active anode (ROS production) coupled to a carbon cathode (HO^•^ production via the EF reaction). The EF/AO combination might enhance both the reaction kinetics and mineralization efficiency while reducing energy consumption and treatment duration.

Therefore, the selection of electrodes is essential for the electrochemical process effectiveness. In this study, carbon felt was chosen as a cathode due to its high electronic conductivity, favorable porosity for reactant accessibility, and mechanical stability, which contributes to its cost-effectiveness. Moreover, it demonstrated good performances in EF of various refractory compounds [27]. Two materials were selected for the anode (active Ti/Pt and non-active Ti_4_O_7_) to investigate different EAOPs (EF, AO, and EF/AO). This study is important because it compared the effectiveness of three EAOPs to address the urgent need for efficient and sustainable methods for toxic dye degradation in wastewater treatment and also because it screened many performance indicators. The impact on the degradation performance of different operational parameters, including current density and initial RhB concentration, was evaluated. Compressed air was used instead of oxygen based on cost-effectiveness, reduced maintenance requirements, and operational simplicity. Toxicity was assessed using a *Vibrio fischeri* bioluminescence inhibition test to obtain a quantitative measurement of the treated dye sample detoxification that was correlated with by-product characterization by Ultra-High Performance Liquid Chromatography coupled with High-Resolution Mass Spectrometry (UHPLC-HRMS) and mineralization rate quantification (from total organic carbon, TOC, measurement). In this way, a direct link could be demonstrated between the degradation of pollutants and their measurable effects on living organisms, consequently enhancing the ecological relevance of our results. Unlike previous research, this study offers an integrated approach that combines optimized electrode selection with rigorously defined operational conditions, toxicity evaluation, and degradation mechanisms.

## 2. Results and Discussion

### 2.1. Current Density Effect

RhB electro-degradation was studied using an initial concentration of 0.01mM RhB (4.79 mg L^−1^) and 50 mM Na_2_SO_4_ as supporting electrolytes at pH 3.0. For the EF experiments, a Ti/Pt anode and a carbon felt cathode were used in the presence of 0.2 mM Fe^2^⁺ as catalysts with air sparging. For the AO experiments, a Ti_4_O_7_ anode and a carbon felt cathode were used under nitrogen sparging without added catalyst. For the EF/AO experiments, a Ti_4_O_7_ anode and a carbon felt cathode were used with 0.2 mM Fe^2^⁺ and air sparging. As current density is a critical parameter that influences both the operational cost and efficiency of electrochemical processes [28], its effect on RhB degradation kinetics by EF, AO, and EF/AO was assessed using a range of current densities (10, 20, 30, 40 and 50 mA cm^−2^). The results (RhB solution absorbance change at λ_max_ = 556 nm) are presented in Figure 1.

Overall, RhB degradation rate increased with increasing current densities (from 10 to 50 mA cm^−2^). At the current density of 30 mA cm^−2^, more than 90% of discoloration, determined at λmax= 556 nm, was achieved within 90 min (EF) or within 30 min (AO and EF/AO). The longer degradation time with EF alone may be attributed to the formation of different intermediate by-products that require prolonged oxidation for complete discoloration. A detailed characterization of these intermediates and their time-dependent formation will be presented in Section 2.6.

To quantify RhB degradation kinetics by EF, AO, and EF/AO, pseudo-first-order rate constants were determined at all current densities applied (10, 20, 30, 40, and 50 mA cm^−2^) (Table 2).

All constant kinetics tended to increase with the current density. This indicates that increasing the current density directly enhances the RhB degradation rate, likely due to an increase in ROS production with all processes. The discoloration kinetics were significantly higher with AO and EF/AO than with EF, regardless of the current density applied. However, at higher current densities, additional factors may start to limit the degradation rate increase, including limitations in the efficiency of H_2_O_2_ generation or the prevalence of competing side reactions that consume reactive species in solution.

The difference in discoloration kinetics could be partly explained by different degradation mechanisms between the EF and AO and EF/AO processes. Moreover, EF/AO involves additional reactions and probably competitive consumption of reactive species, particularly in the first period (0–20 min) of treatment. Nevertheless, both AO and EF/AO displayed similar performance from 30 min onwards.

Importantly, the constant kinetics were estimated from the RhB solution absorbance changes during the treatment. However, this must not be the only parameter taken into account. The study of by-product formation and solution toxicity changes over time are also crucial when determining the process performance (see Section 2.3 and Section 2.6).

Concerning energy consumption, the current density of 30 mA cm^−2^ emerged as optimal for all three processes, offering a favorable balance between a high degradation rate and low energy consumption. Higher current densities might increase the degradation rates but also lead to increased energy consumption and may also promote competing side reactions, such as water formation (Equation (5)), dihydrogen evolution (Equation (6)), and H_2_O_2_ decomposition (Equation (7)) at the cathode due to the higher cell potential.(5)$$O_{2\;}+4H^{+}+4e^{-}\to 2H_{2}O$$(6)$$2H^{+}+2e^{-}\to H_{2}$$(7)$$H_{2}O_{2}+2H^{+}+2e^{-}\to 2H_{2}O$$

### 2.2. Effect of RhB Concentration

To investigate the influence of the initial RhB concentration on the degradation kinetics for the three processes, experiments were performed at a constant current density of 30 mA cm^−2^ using two different RhB concentrations: 0.01 mM (4.79 mg L^−1^) and 0.1 mM (47.9 mg L^−1^). These concentrations are in the same range as the EC50 and LD50 found in the literature, respectively [9].

Analysis of these results (Figure 2) revealed a significant influence of the initial RhB concentration on the degradation kinetics with all three electrochemical treatment methods. This concentration dependence is a key factor for optimizing these processes for practical wastewater applications. A longer electrolysis time to achieve the same level of degradation was required with 0.1 mM RhB than with 0.01 mM RhB. With the same energy supply conditions, the higher initial pollutant load increases the demand for reactive species generation (hydroxyl radicals for EF, ROS for AO, and HO^•^ and ROS for EF/AO), which can be achieved by increasing the electrolysis time.

Nevertheless, even with 0.1 M RhB concentration, the degradation kinetics for all processes still followed a pseudo-first-order model (Table 3). These results, obtained using a carbon-felt cathode in all processes, confirmed the concentration-dependent kinetic degradation (Figure 2). A 10-fold increase in the initial RhB concentration (from 0.01 to 0.1 mM) resulted in a four- to six-fold decrease in the pseudo-first-order constant value, depending on the method. This highlights the effect of the initial substrate concentration on the electrochemical oxidation rate. At the higher RhB concentration, the degradation performance was again higher with AO and EF/AO than with EF [29]. The substantially higher rate constants observed for AO and EF/AO demonstrate the Ti_4_O_7_ anode effectiveness in accelerating the degradation process compared with EF. This is attributed to the superior ability of the Ti_4_O_7_ anode to generate ROS compared with the Ti/Pt anode used in the EF process. Nonetheless, the carbon felt cathode’s high porosity and conductivity contribute to the overall performance of the three processes by facilitating the efficient mass transport and electron transfer for the EF reaction.

### 2.3. Toxicity Monitoring During RhB Degradation

The evaluation of the dye solution toxicity during RhB degradation provides important insights into the toxicological effects of dye effluents in various treatment conditions. In this study, the luminescence inhibition of the marine bacterium *V. fischeri*, an established bioindicator for ecotoxicological assessments, was used to monitor the RhB solution toxicity over time [30,31,32]. The experiments were carried out with initial dye concentrations of 0.01 and 0.1 mM and at a current density of 30 mA cm^2^.

For all the processes and RhB concentrations tested, the dye solution toxicity decreased during the treatment, indicating that the degradation products are all less toxic than RhB (Figure 3). Bioluminescence measurements after 5 and 15 min of exposure to the dye solution samples collected at different time points during the processes provided insights into the immediate toxic effects of RhB and its by-products. The uncertainty analysis, quantified by the coefficient of variation (CV%), underscores the reliability of the toxicity results. The CV% values (*n* = 5) obtained were 12% for inhibition levels below 20%, 5% for inhibition levels between 20% and 70%, and 0% for inhibition levels exceeding 70%, reflecting a rigorous statistical approach [33]. Bioluminescence inhibition changes differed in the function of the degradation processes and the initial dye concentration, suggesting variations in the by-products (nature, concentration) formed during electro-degradation. At the lower RhB concentration, EF/AO was more effective in reducing the dye solution toxicity throughout the duration of the experiment. At the higher RhB concentration, AO led to the formation of less toxic intermediates between 60 and 360 min.

Correlating the dye solution toxicity during the degradation process with the exact nature of the molecules present (dye and by-products) is essential for the comprehensive environmental assessment of the treatment processes. The formed by-products are described in Section 2.6 and Section 2.7, which are dedicated to the identification of the degradation pathways.

### 2.4. Total Organic Carbon Removal Efficiency

TOC removal from the dye solution was assessed by monitoring its temporal changes throughout each 480 min treatment, carried out at constant current density (30 mA cm^−2^) and with an initial RhB concentration of 0.1mM (Figure 4). Overall, TOC removal percentages were higher during EF/AO and AO than EF (CV = 5% of TOC). The best mineralization was achieved by combining EF and AO. This indicated that hydroxyl radicals (HO^•^) and various other ROS, physisorbed on the Ti_4_O_7_ anode surface, play a major role in the oxidative degradation of organic pollutants [34,35,36]. Additionally, the cooperative interaction between AO and Fenton processes facilitated by the carbon-felt cathode enhanced the overall mineralization dynamics. This is in line with the existing literature on the effectiveness of radical species in AOPs for degrading recalcitrant organic compounds [37]. Consequently, coupling these processes optimizes the mineralization rates and improves the overall efficiency of the electrochemical degradation of organic contaminants.

### 2.5. Mineralization Current Efficiency (MCE) and Energy Consumption (EC)

The MCE (Equation (15) in Section 3) was evaluated using 0.1 mM of RhB and a current density of 30 mA cm^−2^ (Figure 5). The highest MCE at the beginning of the electro-oxidation processes was observed with EF/AO due to the increased radical species production on both electrode compartments, promoting efficient RhB oxidation. Regardless of the process or anode materials, MCE gradually decreased over time. This can be attributed to several factors: RhB concentration reduction that decreases the mass transport dynamics, competing side-reactions, formation of degradation-resistant by-products, and decrease in reactive species, such as HO^•^/M(HO^•^), due to scavenger formation that competes with oxidation. The observed MCE values and their decline are in agreement with previous studies showing that initially, MCE values are often high due to rapid radical generation but decrease as treatment progresses due to similar challenges in oxidation and by-product formation [38,39].

The presented results demonstrate the high efficiency of these EAOPs for RhB degradation in controlled conditions. The complexity of real wastewater, with its variable organic load and diverse pollutants, will undoubtedly influence the degradation performance. Previous studies confirmed the effectiveness of these processes for treating complex matrices, underlining their potential for practical applications [40,41].

These findings are particularly relevant for the textile industry, a major cause of dye-contaminated effluents, particularly in developing countries where the problem of treating effluents at the source is crucial due to the possible absence of sewerage systems. Furthermore, these electrochemical technologies are also applicable to the tertiary treatment of domestic wastewater following membrane bioreactor treatment [42].

EC (Equation (17) in Section 3) changes during EF, AO, and EF/AO of 0.1 mM RhB at a constant current density of 30 mA cm^−2^ are presented in Figure 6. EC significantly increased during the treatment until 360 min. This is correlated to the TOC removal increase (Figure 4) and MCE decrease (Figure 5) (with less organic matter, mass transport is reduced). The EC values at 360 and 480 min were similar for the EF and AO processes, whereas the energy requirement was lower for EF/AO. This is explained by the enhanced mineralization efficiency within the same timeframe while all processes maintain a stable cell voltage of 16 V. Overall, EC decreased progressively from EF to AO to EF/AO. EF required ~12 kWh g^−1^ TOC for 68% TOC removal; AO used 9 kWh g^−1^ TOC for 83% TOC removal, and EF/AO needed only 7 kWh g^−1^ TOC for 90% TOC removal. These EC values are among the lowest reported [43], highlighting the high process efficiencies. This study underscores the strategic application of high dioxygen overpotential anodes (e.g., Ti_4_O_7_) in AOPs for efficient but non-selective organic pollutant removal while minimizing the energy requirement.

### 2.6. RhB and By-Product Identification by UHPLC-HRMS

RhB and its degradation by-products were identified in dye solution samples collected at specific time intervals (0, 60, 240, and 480 min for EF and 0, 30, 120, and 360 min for EF/AO) (0.1 mM RhB initial concentration), using UHPLC-HRMS. The identification parameters are summarized in Table 4. Analysis of the chromatograms revealed the appearance of *N*-desethylated intermediates during the electrochemical treatment, consistent with the literature that recognizes these specific intermediates as common products of RhB degradation [44]. The RhB peak intensity progressively decreased, accompanied by an increase in the intensity of the intermediate peaks, followed by their subsequent reduction over time. Similar by-products were formed during EF and EF/OA but significantly later during EF. Indeed, the RhB peak had completely disappeared after 480 min of EF and only 120 min of EF/AO (Figure 7), in agreement with the gradual discoloration of the dye solution.

Moreover, the changes in the compounds present in the dye solution (through the analysis of the peak intensity of RhB and its by-products) were correlated with the decrease of the solution toxicity (see Section 2.4). This elucidates the correlation between RhB degradation and toxicity attenuation, confirming the treatment effectiveness.

### 2.7. Possible RhB Degradation Pathways

ROSs play a critical role in the degradation of organic molecules, such as dyes, by inducing the cleavage of specific bonds, leading to their breakdown [45]. This cleavage occurs through several mechanisms: (i) hydrogen abstraction, where hydroxyl radicals (HO^•^) remove hydrogen from carbon-containing bonds (C–H) to form carbon-centered radicals that destabilize the molecule [46]; (ii) addition reactions where ROSs combine with double bonds (C=C), resulting in new radical formation that prompts further degradation [47]; and (iii) peroxidation, in which compounds react with peroxides (i.e., H_2_O_2_) to create peroxy radicals (R–O–O^•^) that can repeat the process of bond cleavage [48,49]. Dyes (e.g., xanthenes) can also undergo oxidative degradation because their structures contain double bonds and functional groups that can be oxidized by ROSs [50]. For instance, hydroxyl radicals can attack double bonds in the xanthene structure, causing decolorization and breakdown into less toxic products. After bond cleavage, the resulting radicals may undergo additional reactions, ultimately resulting in the complete mineralization of organic compounds into carbon dioxide, water, and other simple molecules.

On the basis of the literature, two main RhB degradation pathways can be proposed (Figure 8). Active radical species, such as hydroxyl (HO^•^), may attack the RhB molecules through immediate *N*-desethylation (Pathway 1), leading to the formation of nitrogen-centered radicals that facilitate the degradation of the RhB chromophore structure [51,52,53]. Alternatively or concomitantly, RhB can first undergo decarboxylation followed by *N*-desethylation (Pathway 2).

The mass spectral interpretation of our UHPLC-HRMS data (Figure 7 and Table 4) allowed identifying the following intermediate by-products: *N*,*N*-diethyl-*N*′-ethylrhodamine, *N*,*N*-diethylrhodamine, *N*-ethyl-*N*′-ethylrhodamine and *N*-ethylrhodamine, with *m/z* values of 443.47, 415.47, 387.47 and 359.47, respectively. These findings indicate that “Pathway 1” is the predominant mechanism of RhB degradation during EF and EF/AO, with faster by-product formation (and thereby RhB decomposition) during EF/AO. The *N*-desethylated intermediates could then decompose into other intermediates with an *m/z* of 331.47. The inherent instability of the generated fragments likely leads to the formation of additional intermediates with lower *m/z* values. These intermediates could be oxidized to form phthalic acid, phthalic anhydride, benzoic acid, and lactic acid, with *m/z* values of 166.47, 148.47, 122.47, and 90.47, respectively. Ultimately, these intermediates may decompose into final products, including CO_2_ and H_2_O.

Concerning the proposed “Pathway 1”, smaller intermediates with *m/z* ≤ 331.47 could not be identified within the 6 h treatment period. The identification of by-products with low concentration and/or low sensitivity needs pre-concentration steps and optimization of the analytical techniques. This part of the study is currently in progress.

## 3. Materials and Methods

### 3.1. Chemicals

All chemicals employed in this study were of analytical grade, ensuring high purity and reliability for experimental procedures. RhB (Basic Violet 10, chloride of [9-(2-carboxyphenyl)-6-(diethylamino)xanthene-3-ylidene]-diethylammonium) (C_28_H_31_ClN_2_O_3_, >98%) was from Thermo Scientific. Anhydrous sodium sulfate (Na_2_SO_4_, ≥99%) and iron(II) sulfate heptahydrate(FeSO_4_.7H_2_O, ≥99%) were from Sigma-Aldrich (Saint Quentin Fallavier, France) and sulfuric acid (H_2_SO_4_, 95–98%) from Labkem. All solutions were prepared using pure water (10 to 15 MΩ.cm) produced with an Elix Essential system.

### 3.2. Electrochemical Cell

Experiments were performed at ambient temperature in a two-electrode undivided cell system with a total volume capacity of 90 mL. The anode configuration included a Ti/Pt grid or a Ti_4_O_7_ electrode (Saint-Gobain Coating Solutions, Avignon, France). The anode was arranged in parallel alignment to the carbon felt cathode (Alfa Aesar, Illkirch, France), as depicted in Figure 9.

The efficiency of AO, EF, and their combination is markedly influenced by several operational parameters, including the solution pH, the nature of the supporting electrolyte, the concentration of the added catalyst, the amount of H_2_O_2_ generated at the electrode (for EF and EF/AO), the applied current density and the electrode type [54,55,56].

In this study, each experiment was carried out with 80 mL of RhB solution prepared using 50 mM Na_2_SO_4_ as a supporting electrolyte. FeSO_4_·7H_2_O was optionally introduced as a Fenton catalyst at the concentration of 0.2 mM for EF and EF/AO [20].

The solution pH was adjusted to the optimal value of 3.0 [44] using a pH meter (PHM210 MeterLab, Radiometer Analytical) and sulfuric acid (H_2_SO_4_). It is well known that conventional homogeneous EF is carried out at acidic pH values (2.0–4.0) [19,44] because low pH values favor H_2_O_2_ production, stabilization of the catalyst Fe^2+^, and better activity of H_2_O_2_ and HO^•^. Working at a more neutral pH is still a challenge that is addressed in ongoing research (e.g., heterogeneous EF implying cathode modification, chelating agent addition) [57,58].

Different current densities (10, 20, 30, 40, and 50 mA cm^−2^) were investigated using a power supply (ELC DC Power Supply AL 781NX, ELC, Annecy, France). To ensure adequate mass transfer to the electrodes, the solution was continuously stirred with a magnetic stirring bar. Additionally, to enhance oxygen dissolution and its subsequent electrochemical conversion into H_2_O_2_, compressed air was bubbled through the system, starting 15 min before electrolysis. The primary advantage of using compressed air is its cost-effectiveness and reduced maintenance requirements.

### 3.3. Performance Indicators of RhB Degradation and Associated Analytical Methods

Hydroxyl radicals and other ROS produced during EF and/or AO are characterized by their non-selective nature and high reactivity with organic compounds and possess a very short lifespan (nanoseconds for HO^•^ and up to a few seconds for other alkyl radicals) [59]. Consequently, their transient existence excludes their accumulation in the environment, complicating their determination. Therefore, they were not quantified in this study. Instead, the study focused on the impact of the different EAOPs by screening different performance indicators that could qualify and quantify the dye solution composition changes over time.

#### 3.3.1. Color Removal (UV–Visible Spectrophotometry Analysis)

RhB degradation in the dye solutions under treatment was monitored by UV–visible spectrophotometry (Uviline Connect 940 from Secomam, Dustcher, Brumath, France). To check the validity of the Beer–Lambert Law for RhB quantification, five standard solutions of RhB in 50 mM Na_2_SO_4_ (pH 3) were prepared and measured against a blank sample of Na_2_SO_4_ at the wavelength of maximum absorbance (λ_max_ = 556 nm) of RhB, using quartz cuvettes (Hellma Analytics, Fisher Scientific, Illkirch, France). A valid linear calibration equation (R^2^ > 0.9950) was obtained.

The percentage of solution color removal was determined using Equation (8).(8)$$\mathrm{color}\;\mathrm{removal}\;\left(\%\right)=100\;\times \;\frac{({\mathrm{Abs}}_{0}-\mathrm{Abs})}{{\mathrm{Abs}}_{0}}$$
where ${\mathrm{Abs}}_{0}$ and Abs are the initial absorbance and the absorbance at electrolysis time (*t*), respectively, that are linearly proportional to C_0_ (RhB initial concentration) and C (RhB concentration at time *t*), respectively.

#### 3.3.2. RhB Degradation Kinetic Constants

RhB degradation kinetics were quantified by determining the apparent constant K_app_, assuming that the process follows a pseudo-first-order kinetic, as described by Equations (9) and (10).(9)$$\frac{-d\left[C\right]}{\mathrm{dt}}=K_{\mathrm{app}}{\left[C\right]}_{0}$$(10)$$\mathrm{Ln}\;\frac{\left[C\right]}{{\left[C\right]}_{0}}={-K}_{\mathrm{app}}t\;$$
where C_0_ is RhB’s initial concentration, C is the concentration at time *t*, and k is the associated degradation rate constant.

Uncertainties on the K_app_ value during EF were assessed by performing triplicate experiments in identical conditions (30 mA cm^−2^), yielding a coefficient of variation (CV%) of 10% at most, indicating satisfactory experimental repeatability. For the other conditions studied (other current densities and processes), the CV% was estimated by statistical analysis of each linear regression that depends on the number of experimental points taken into account in the linear relationship and the regression quality (R^2^ value).

#### 3.3.3. Toxicity Test

The bacterial toxicity assessment was determined with a Microtox^®^ Model 500 Analyzer (Modern Water Inc., York, UK) in accordance with the international standard procedure ISO 11348-3 [60]. The toxicity of RhB and its intermediate degradation by-products was evaluated by measuring the inhibition of bioluminescence in the marine bacterium *V. fischeri*. The bioluminescence emission is intrinsically linked to cellular respiration, providing a direct correlation with cellular activity [20]. The bioluminescence inhibition I(t) (%) was calculated after 5 and 15 min of exposure to the dye solution samples, using Equation (11). A screening test of 81.9% was used to characterize inter-sample toxicity variability and identify the relative toxicity of each sample solution. A 22% NaCl solution was employed to allow *V. fischeri* normal activity and, thus, luminescence emission.(11)$$I\;\left(t\right)\;\%=\left(1-\frac{\mathrm{LU}\;\left(t\right)}{\mathrm{LU}\;\left(0\right)}\right)\times 100\;\;$$
where LU(0) represents the initial bacterial bioluminescence intensity before the addition of the dye sample. LU(t) is the bacterial bioluminescence intensity after 5 or 15 min of exposure to the dye solution. In the absence of toxicity, bacterial bioluminescence naturally diminishes over time. Therefore, errors due to the luminescence R(t) variability in the control solution (MilliQ water and NaCl), which gives the LU(0) values, need to be compensated using Equation (12).(12)$$R\;\left(t\right)=\frac{{\mathrm{LU}}_{0\;}\;(t)}{{\mathrm{LU}}_{0}\;(0)}\;$$
where LU_0_ (0) is the initial bacterial bioluminescence intensity before the addition of the control solution, and LU_0_ (t) is the bioluminescence intensity after 5 and 15 min of contact with the control solution. Then, Equation (13) describes the corrected inhibition rate attributed to the sample toxicity:(13)$$\mathrm{Ic}\;\left(t\right)\;\%=\left(1-\;\frac{\mathrm{LU}\;\left(t\right)}{R\;\left(t\right)\times \;\mathrm{LU}\;\left(0\right)}\right)\times \;100\;$$

#### 3.3.4. Total Organic Carbon (TOC) Removal

RhB mineralization in the solutions was monitored through TOC measurements using a TOC-L CSH/CSN analyzer (Shimadzu, Marne-La-Vallée, France). Calibration curves were built using automatic dilutions of a TOC standard solution (potassium hydrogen phthalate). The TOC removal efficiency was calculated using Equation (14).(14)$$\%\;\mathrm{TOC}\;\mathrm{Removal}\;\mathrm{efficiency}=\left(\frac{{\mathrm{TOC}}_{0}-{\mathrm{TOC}}_{t}}{{\mathrm{TOC}}_{0}}\right)\times \;100\;\;$$
where TOC_0_ and TOC_t_ represent the TOC concentration at the beginning of the experiment and at different electrolysis time points (*t*) (mg of carbon L^−1^), respectively.

The TOC values were used to calculate the MCE according to Equation (15) [61].(15)$$\mathrm{MCE}\;\left(\%\right)=\left(\frac{\Delta {\left(\mathrm{TOC}\right)}_{\exp}\;n\;F{\;V}_{s}}{4.32\;\times \;10^{7}\;m\;I\;t}\right)\times \;100$$
where ∆(TOC)_exp_ is the change in experimental TOC (mg carbon L^−1^), n is the number of electrons involved in RhB mineralization, F is the Faraday constant (96,485 C mol^−1^), V_s_ is the solution volume (L), 4.32 × 10^7^ is a conversion factor (3600 s h^−1^ × 12,000 mg C mol^−1^), m is the number of carbon atoms in RhB (28 atoms), I is the current applied (A), and t represents the electrolysis duration (h). For RhB mineralization, the number of electrons (n) is 130, according to Equation (16), because RhB converts to CO_2_ and NH_4_^+^ [62].C_28_H_31_N_2_O_3_^+^ + 53 H_2_O → 28 CO_2_ + 2 NH_4_^+^ + 129 H^+^ + 130 e^−^(16)

TOC results were also used to determine the EC, expressed as Current Efficiency (kWh g^−1^ TOC), with Equation (17) [63].(17)$$\mathrm{EC}=\frac{E_{\mathrm{cell}}\;I\;t}{\Delta {\left(\mathrm{TOC}\right)}_{\exp}V_{s}}$$
where E_cell_ is the average cell voltage (V), I is the current density (A), t is the electrolysis duration (h), ∆ (TOC)_exp_ is the observed TOC decrement (mg C L^−1^), and V_s_ is the volume of treated solution (L).

#### 3.3.5. RhB and By-Product Identification (UHPLC-HRMS Analysis)

RhB degradation products were identified using UHPLC-HRMS. The UHPLC-HRMS system was a Vanquish model (Thermo Fisher Scientific, Illkirch, France) integrated with a SYNAPT G2-S mass spectrometer (Waters Corporation, Manchester, UK) featuring an Electrospray Ionization source.

The UHPLC-HRMS separation was carried out using a Waters Atlantis Premier BEH C18 AX column (1.7 µm × 2.1 × 50 m) (Waters S.A.S., Saint-Quentin-en-Yveline, France) and the following mobile phases (A: 0.1% formic acid in water; and B: 0.1% formic acid in acetonitrile). A gradient elution was performed, starting with 100% phase A from 0 to 3 min and transitioning to 100% phase B from 3 to 5 min. The flow rate was set at 0.5 mL/min, with an injection volume of 10 µL, and the column temperature was maintained at 25 °C.

High-resolution electrospray ionization-mass spectrometry was performed in positive or negative ion mode. The detection conditions were capillary voltage of 3000 V, cone voltage of 20 V, dry gas temperature of 140 °C, desolvation temperature of 450 °C, dry gas flow rate of 1000 L/h, and nitrogen as nebulizing gas at a pressure of 6.5 bar. Leucine enkephalin (1 ng/µL) was used as an internal standard for calibration.

## 4. Conclusions

This study showed efficient RhB degradation using EAOPs. Increasing the current density significantly accelerated RhB degradation, and 30 mA cm^−2^ was optimal by balancing energy consumption and degradation efficiency. The performance of EF, AO, and their combination (EF/AO) was quantified, revealing that all could efficiently remove RhB by converting the dye into significantly less toxic by-products. RhB concentration influenced the degradation kinetics: higher concentrations required longer treatment times. Toxicity monitoring, using the *V. fischeri* bioluminescence inhibition test and TOC measurements, highlighted drastic toxicity reduction (60–90%) and pollutant mineralization (68–90%) during RhB electrochemical degradation, depending on the process applied and/or the initial dye concentration used. At low RhB concentration, EF/AO exhibited superior performance, followed by AO and EF. Estimation of the MCE and EC values also indicated that EF/AO was the best-performing technology for RhB electro-degradation. UHPLC-HRMS confirmed that the RhB degradation pathway included *N*-desethylation and subsequent oxidation reactions, resulting in the formation of less toxic intermediate products and their mineralization to CO_2_ and H_2_O.

To our knowledge, previous studies never investigated at the same time RhB degradation using different treatment processes and different analytical parameters. This study addresses this gap by providing an exhaustive assessment of the degradation of this refractory dye using different EAOPs and by screening a panel of performance indicators (i.e., color removal, TOC, energy efficiency, by-product identification, degradation mechanisms, and toxicity for every treatment process). Therefore, this work offers crucial insights for addressing the issue of RhB pollution in waters and confirms the huge potential of EAOPs as efficient, sustainable, and environmentally friendly wastewater treatment technologies.

## Figures and Tables

**Figure 1 molecules-30-00712-f001:**
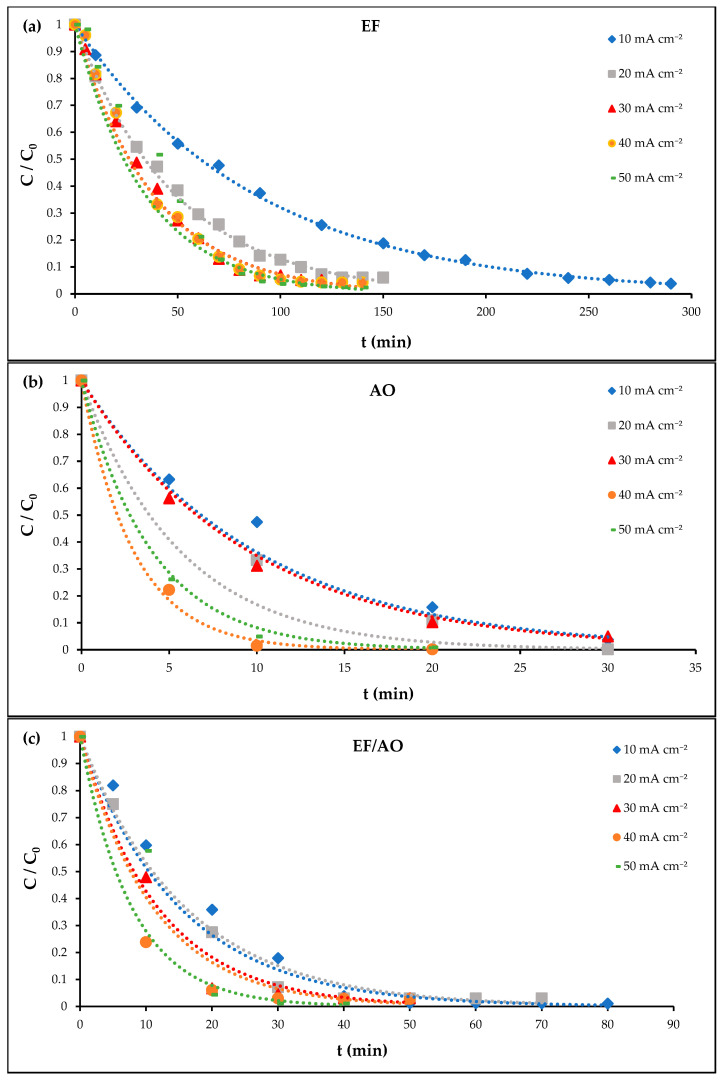
Influence of current density on RhB degradation, measured at λ_max_ = 556 nm, employing a carbon felt (CF) cathode and a Ti/Pt or a Ti_4_O_7_ anode. Experimental conditions were as follows: [RhB] = 0.01 mM in 50 mM Na_2_SO_4_ at pH 3.0; (**a**) EF alone (CF + Ti/Pt electrodes) with 0.2 mM Fe^2^⁺ and air sparging; (**b**) AO alone (CF + Ti/Pt electrodes) with N_2_ sparging in the absence of the Fe^2+^ catalyst; (**c**) EF/AO (CF + Ti_4_O_7_ electrodes) with 0.2 mM Fe^2^⁺ and air sparging.

**Figure 2 molecules-30-00712-f002:**
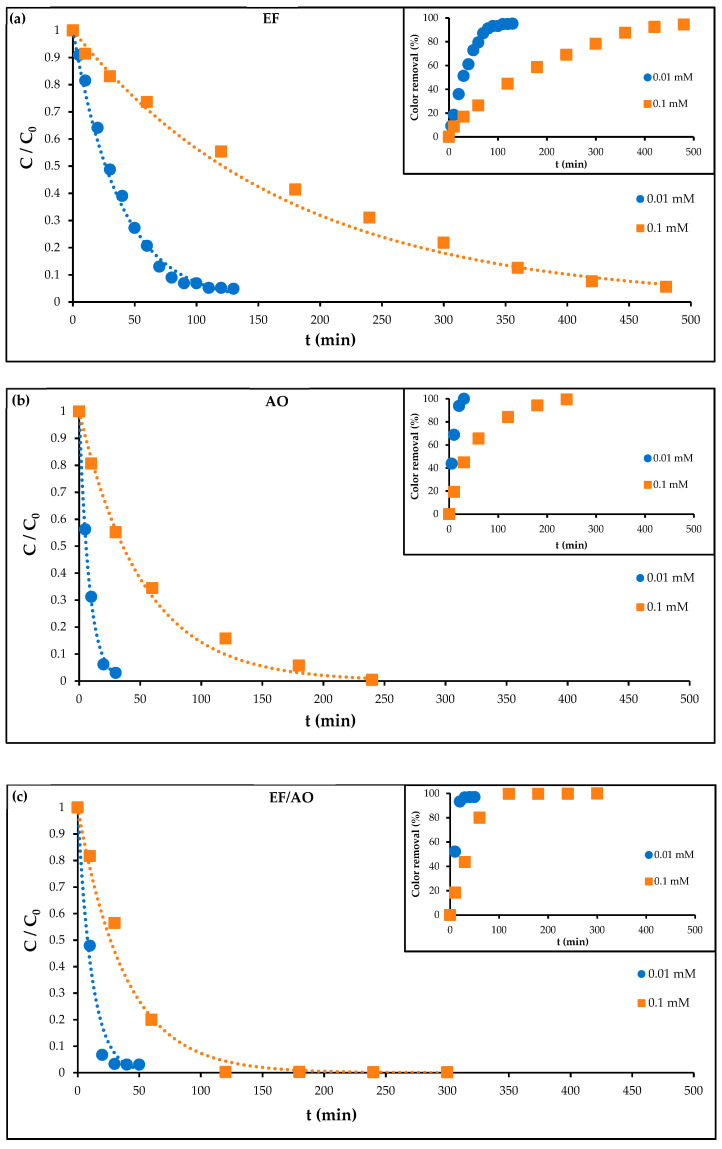
RhB degradation kinetics at the initial concentrations of 0.1 mM and 0.01 mM in the following conditions: 50 mM Na_2_SO_4_, pH 3.0 and I = 30 mA cm^−2^, using (**a**) EF with 0.2 mM Fe^2^⁺ and air sparging; (**b**) AO with N_2_ sparging; (**c**) EF/AO with 0.2 mM Fe^2^⁺ and air sparging.

**Figure 3 molecules-30-00712-f003:**
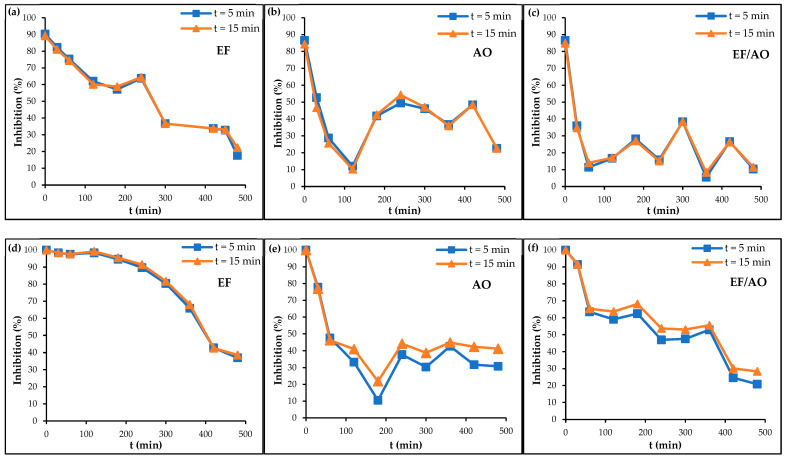
Changes in *V. fischeri* bioluminescence inhibition during the degradation of 0.01 mM RhB using the (**a**) EF, (**b**) AO, and (**c**) EF/AO processes and of 0.1 mM RhB using the (**d**) EF, (**e**) AO and (**f**) EF/AO processes.

**Figure 4 molecules-30-00712-f004:**
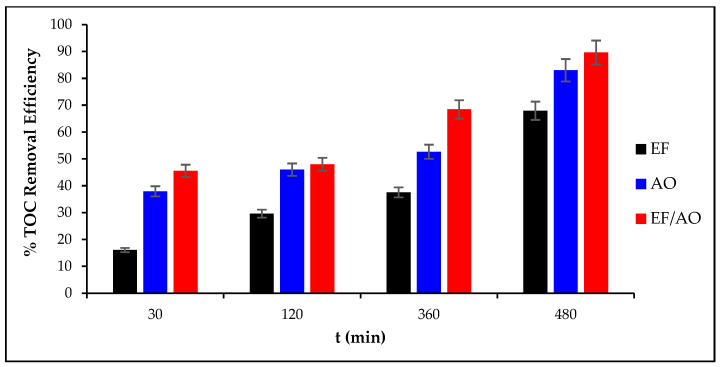
Mineralization of 0.1 mM RhB by the EF, AO, and EF/AO processes (480 min of treatment at 30 mA cm^−2^).

**Figure 5 molecules-30-00712-f005:**
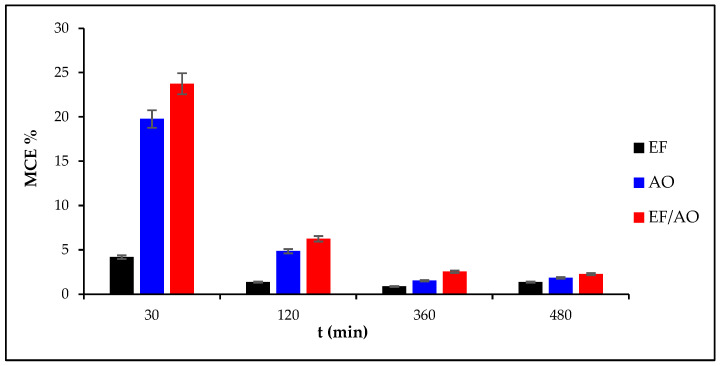
Mineralization Current Efficiency (MCE) of 0.1M RhB in the function of the electrolysis time at a current density of 30 mA cm^−2^.

**Figure 6 molecules-30-00712-f006:**
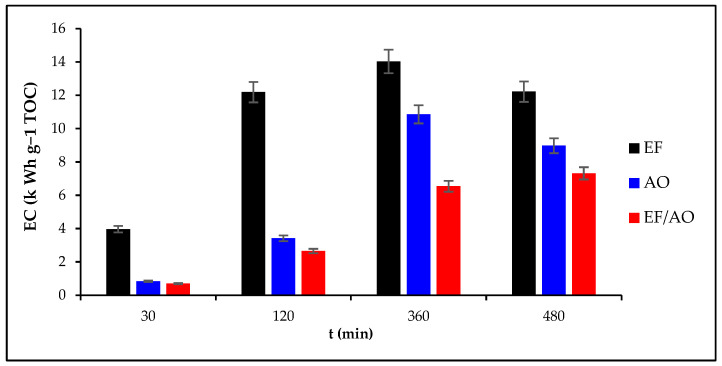
Energy consumption (EC) for the removal of 0.1 mM RhB in the function of the electrolysis duration with a current density of 30 mA cm^−2^ for the EF, AO, and EF/AO processes.

**Figure 7 molecules-30-00712-f007:**
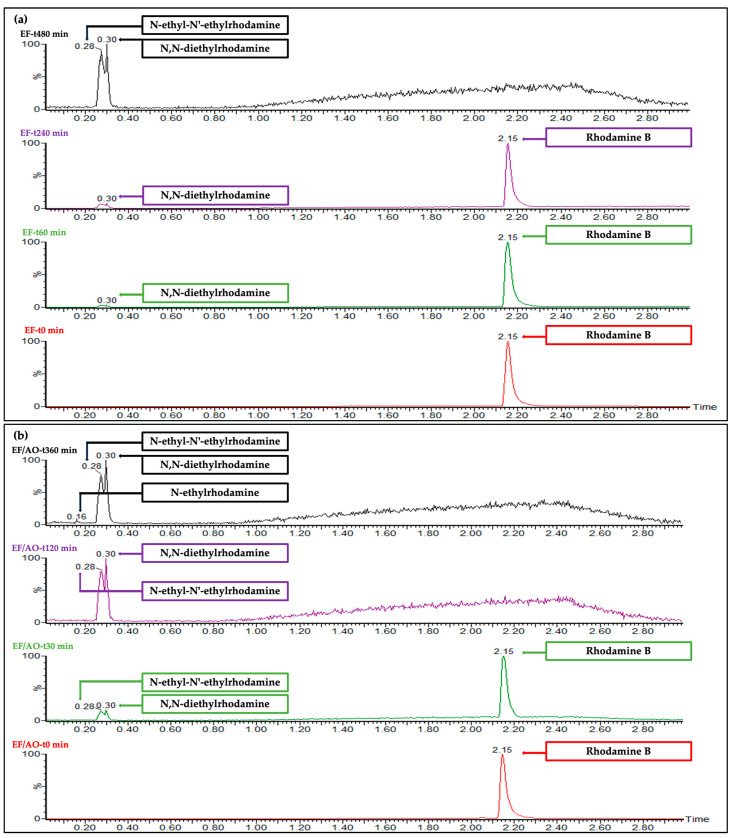
UHPLC-HRMS chromatograms of 0.1 mM RhB obtained at the indicated time points during the (**a**) EF and (**b**) EF/AO processes.

**Figure 8 molecules-30-00712-f008:**
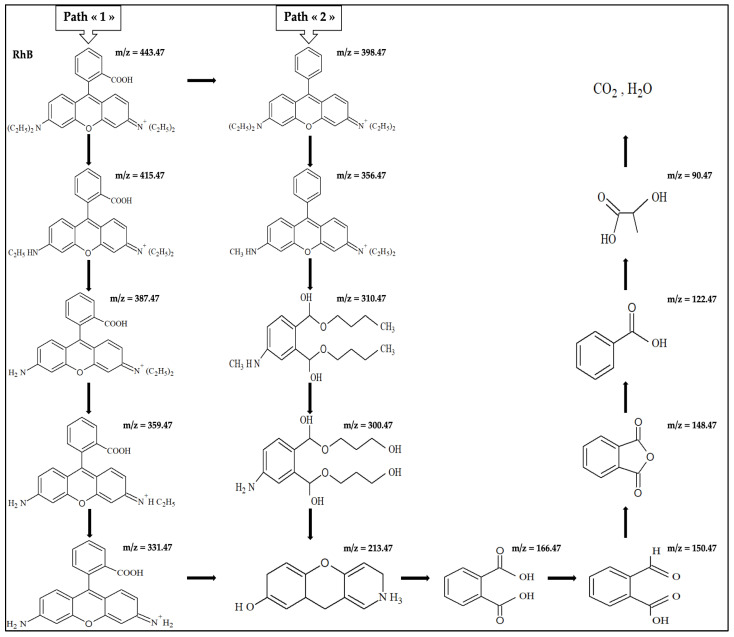
Investigation of the possible degradation pathways for 0.1 mM RhB by EF/AO at the current density of 30 mA cm^−2^.

**Figure 9 molecules-30-00712-f009:**
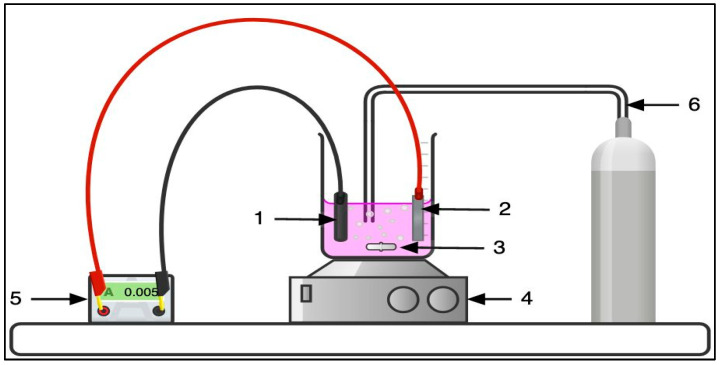
Schematic representation of the experimental setup: (1) Carbon-felt cathode, (2) Anode (Ti/Pt or Ti_4_O_7_), (3) Magnetic stirring bar, (4) Magnetic stirrer, (5) Power supply, (6) Compressed air.

**Table 1 molecules-30-00712-t001:** Molecular structure and physicochemical characteristics of rhodamine B.

Molecular Structure	Physicochemical Characteristics	
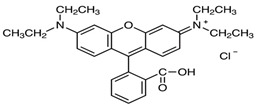	Chemical formula of RhB as a chloride salt	C_28_H_31_ClN_2_O_3_
	Molecular weight	479.01 g/mol
	Class	Xanthene
	Maximum absorption wavelength	546–556 nm

**Table 2 molecules-30-00712-t002:** Apparent pseudo-first-order rate constants (k_app_) for RhB degradation by EF, AO, and EF/AO in the presence of 50 mM Na_2_SO_4_ at pH 3.0 and various current densities (from 10 to 50 mA cm^−2^).

Process	Current Density (mA cm^−2^)	k_app_ (min^−1^)	R^2^
EF	10	0.0111 ± 0.0011	0.9983
	20	0.0206 ± 0.0021	0.9947
	30	0.0277 ± 0.0028	0.9804
	40	0.0284 ± 0.0028	0.9863
	50	0.0293 ± 0.0059	0.9288
AO	10	0.0703 ± 0.0070	0.9942
	20	0.1100 ± 0.0110	1.0000
	30	0.1340 ± 0.0134	0.9880
	40	0.2152 ± 0.0430	0.9785
	50	0.2501 ± 0.0500	0.9732
EF/AO	10	0.0448 ± 0.0090	0.9538
	20	0.0872 ± 0.0087	0.9914
	30	0.1127 ± 0.0225	0.9500
	40	0.1409 ± 0.0282	0.9998
	50	0.1519 ± 0.0304	0.9988

**Table 3 molecules-30-00712-t003:** Apparent pseudo-first-order rate constants (k_app_) for RhB degradation (0.01 and 0.1 mM) by EF, AO, and EF/AO: 50 mM Na_2_SO_4_; pH 3.0; I = 30 mA cm^−2^ (+ Fe^2^⁺ = 0.2 mM in EF and EF/AO).

Dye	Process	Anode	Cathode	RhB Initial Concentration (mM)			
				0.01		0.1	
				k_app_ (min^−1^)	R^2^	k_app_ (min^−1^)	R^2^
RhB	EF	Ti/Pt	CF	0.0277	0.9804	0.0057	0.9846
	AO	Ti_4_O_7_	CF	0.1340	0.9880	0.0210	0.9647
	EF/AO	Ti_4_O_7_	CF	0.1127	0.9500	0.0252	0.9701

**Table 4 molecules-30-00712-t004:** UHPLC-HRMS identification parameters (t_R_ and *m*/*z*) of RhB and the aromatic by-products generated during the EF and EF/AO processes.

Identification Parameters		
*m*/*z*	t_R_ (min)	Molecule
443.47	2.15	Rhodamine B ion
415.47	0.30	*N*,*N*-diethylrhodamine
387.47	0.28	*N*-ethyl-*N*′-ethylrhodamine
359.47	0.16	*N*-ethylrhodamine

## Data Availability

The original contributions presented in this study are included in the article. Further inquiries can be directed to the corresponding author.

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
