# Peer review of "Screening Refractory Dye Degradation by Different Advanced Oxidation Processes"

_molecules, 2025, doi:10.3390/molecules30030712_

Round 1

Reviewer 1 Report

Comments and Suggestions for Authors

The manuscript can be accepted after the following revisions,

1. The pH effects on the degradation performance in the three advanced oxidation processes should be investigated.

2. If and how the other common pollutants present in wastewater affect the degradation peroformance in the three advanced oxidation processes?

3. The real watsewater samples are strongly suggested to be tested in the three advanced oxidation processes.

4. The degradation pathways should be more comprehensive in the three advanced oxidation processes. how the ROS reacted with the pollutants? which ROS species mainly contribute to the great degradation performance?

Author Response

Comment 1: The pH effects on the degradation performance in the three advanced oxidation processes should be investigated.

Response 1: The question of pH is indeed a fundamental one in liquid effluent treatment processes, and more particularly in aqueous effluents, the target of our study. Nevertheless, the aim of our study was to compare three advanced electrochemical oxidation processes: electro-Fenton (EF), anodic oxidation (AO) and EF/AO coupling. The EF process in homogeneous phase only operates at acid pH (optimum pH 3), so for the purposes of process comparison, this is the pH value we set. It is recognized, however, that the heterogeneous-phase EF process can operate at a neutral pH but this choice is still a challenge that is addressed in ongoing research (e.g. heterogeneous EF implying cathode modification, chelating agent addition) [new 57,58]. For this reason, the effect of pH on each of the processes can only be considered in the future in the case of heterogeneous-phase EF.

It should be noted that pH may also have an influence on the physico-chemistry of Rhodamine, but we did not wish to investigate this aspect so as not to interfere with the comparison of the performance of each process.

Better justifications for our choice of pH are now specified in lines 381 to 386 of the revised document, as follows: “It is well known that conventional homogeneous EF is carried out at acidic pH values (2.0 - 4.0) [19,44], because low pH values favor H2O2 production, stabilization of the catalyst Fe2+ and better activity of H2O2 and HO•. Working at a more neutral pH is still a challenge that is addressed in ongoing research (e.g. heterogeneous EF implying cathode modification, chelating agent addition) [new 57,58]”.

Comment 2: If and how the other common pollutants present in wastewater affect the degradation peroformance in the three advanced oxidation processes?

Response 2: This is an important question, since these processes are intended to be applied to natural and/or industrial effluents, which are always complex. In a previous paper, we have already shown that it is possible, for example, to efficiently degrade 100 ppb carbamazepine in an effluent containing 10 ppm of organic carbon [new 40].

Advanced oxidation is mediated by radical species with very low selectivity. This is why the total quantity of organic matter present in the effluent (in the form of natural or synthetic organic matter) may influence the degradation performance of the target molecule (because organic matter will also consume radical species). Nevertheless, these advanced oxidation processes are very powerful and modular (e.g. by adjusting current density to increase radical formation), while however being attentive to the possible formation of AOX, perchlorate and chlorate in chloride medium.

This point has been clarified in lines 269 to 272 of the document as follows: “The complexity of real wastewater, with its variable organic load and diverse pollutants, will undoubtedly influence the degradation performance. Previous studies confirmed the effectiveness of these processes for treating complex matrices, underlining their potential for practical applications [new 40, new 41]”.

Comment 3: The real watsewater samples are strongly suggested to be tested in the three advanced oxidation processes.

Response 3: The objectives of our study are already multiple: i) to compare 3 treatment processes for RhB degradation under well controlled operating conditions, ii) to study several treatment performance indicators, iii) to elucidate the RhB degradation pathway. This is why we chose a simple RhB solution rather than a complex one, in order to fully meet our objectives. Moreover, this question is indeed directly related to the previous one, and we have already been able to study this question in our laboratory, but for another subject (the case of EF treatment of landfill leachate) [new 41].

In order to consider the referee’s comments (both questions 2 and 3), we have added new comments and references [40,41] in line 269-272: “The complexity of real wastewater, with its variable organic load and diverse pollutants, will undoubtedly influence the degradation performance. Previous studies confirmed the effectiveness of these processes for treating complex matrices, underlining their potential for practical applications [new 40, new 41]”.

Comment 4: The degradation pathways should be more comprehensive in the three advanced oxidation processes. how the ROS reacted with the pollutants? which ROS species mainly contribute to the great degradation performance?

Response 4: We consider that the degradation pathways presented in the manuscript are comprehensive enough, with regard to the objectives of our work that follow a more applicative than a fundamental trajectory: Comparison of 3 treatment processes for RhB degradation, through the investigation of multiple treatment performance indicators.

We agree with the referee about the importance of elucidating the action mechanisms of ROS but we think that this research point is out the scope of this work. We already mentioned in the material and methods Section (lines 397-399) why no attempt was made to quantify ROS in this study. This paragraph has now been modified (lines 395-402) to better clarify the applicative orientation of our work as follows: “Hydroxyl radicals and other ROS produced during EF and/or AO are characterized by their non-selective nature and high reactivity with organic compounds, and possess very short lifespan (nanoseconds for HO· and up to few seconds for other alkyl radicals)[59]. Consequently, their transient existence excludes their accumulation in the environment, complicating their determination. Therefore, they were not quantified in this study. Instead, the study focused on the impact of the different EAOPs by screening different performance indicators that could qualify and quantify the dye solution composition changes over time”.

In addition, we already proposed 2 complete degradation pathways of the RhB molecule and we could also identify one predominant path amongst treatment processes explored (lines 337-343): The mass spectral interpretation of our UHLPC-HRMS data (Figure 7 and Table 4) allowed identifying the following intermediate by-products: N,N-diethyl-N'-ethylrhodamine, N,N-diethylrhodamine, N-ethyl-N'-ethylrhodamine and N-ethylrhodamine, with m/z values of 443.47, 415.47, 387.47 and 359.47, respectively. These findings indicate that “Pathway 1” is the predominant mechanism of RhB degradation during EF and EF/AO, with faster by-product formation (and thereby RhB decomposition) obtained for the coupling process…”.

We also provided some explanations about the radical action of reactive species ROS (HO· and other radical species formed in the medium), as shown in lines 331- 336: Active radical species, such as hydroxyl (HO·), may attack the RhB molecules through immediate N-desethylation (Pathway 1), leading to the formation of nitrogen-centered radicals that facilitate the degradation of RhB chromophore structure [51-53]. Alternatively or concomitantly, RhB can first undergo decarboxylation followed by N-desethylation (Pathway 2)”.

Finally, to support the referee’s comment, we have decided to add a new paragraph about the action of ROS on dyes, from literature (new lines: 316 - 330), as follows: “ROS play a critical role in the degradation of organic molecules, such as dyes, by inducing the cleavage of specific bonds, leading to their breakdown [45]. This cleavage occurs through several mechanisms: i) hydrogen abstraction, where hydroxyl radicals (HO•) remove hydrogen from carbon-containing bonds (C-H) to form carbon-centered radicals that destabilize the molecule [new 46]; ii) addition reactions where ROS combine with double bonds (C=C), resulting in new radical formation that prompts further degradation [new 47]; and iii) peroxidation, in which compounds react with peroxides (i.e. H2O2) to create peroxy radicals (R-O-O•) that can repeat the process of bond cleavage [new 48, new 49]. Dyes (e.g., xanthenes) can also undergo oxidative degradation because their structures contain double bonds and functional groups that can be oxidized by ROS [new 50]. For instance, hydroxyl radicals can attack double bonds in the xanthene structure, causing decolorization and breakdown into less toxic products. After bond cleavage, the resulting radicals may undergo additional reactions, ultimately resulting in the complete mineralization of organic compounds into carbon dioxide, water and other simple molecules”.

Reviewer 2 Report

Comments and Suggestions for Authors

The paper entitled “Screening refractory dye degradation by different advanced oxidation processes” reports the degradation of Rhodamine B (RhB) using advanced electrochemical oxidation processes (EAOPs). Additionally, the degradation pathways of RhB was also elucidated via high-resolution mass spectrometry analysis. Therefore, this manuscript is recommended for publication after the following issues are properly addressed.

1.      How about the generality of the method? Can this EAOPs be extended to the degradation of other organic dyes?

2.      The advantages of the EAOPs are recommended to be discussed in compared with other general Fenton oxidation.

3.      The concentration of the RhB should be optimized.

4.      English language and grammar of the manuscript are recommended to be improved.

5.      The format of the references should be checked and revised according to the author’s guideline of the journal.

Comments on the Quality of English Language

None

Author Response

Comment 1: How about the generality of the method? Can this EAOPs be extended to the degradation of other organic dyes?

Reply 1: General information on EAOP methods has been given in lines 60 to 96. Moreover, it is well known that advanced oxidation is mediated by reactive radical species with very low selectivity. This means that radical species are able to attack many compounds, so most natural organic compounds (e.g. humic acids) or synthetic compounds (e.g. dyes, pharmaceuticals, chemicals) will also be degraded.

In response both to Reviewer 1 and Reviewer 2, this point has been clarified in lines 268 to 272 of the document, as follows: “The complexity of real wastewater, with its variable organic load and diverse pollutants, will undoubtedly influence the degradation performance. Previous studies confirmed the effectiveness of these processes for treating complex matrices, underlining their potential for practical applications [new 40, new 41]”.

Comment 2: The advantages of the EAOPs are recommended to be discussed in compared with other general Fenton oxidation.

Reply 2: This point is fundamental to the justification of the study, and it is therefore important to specify the advantages of electrochemical techniques over chemical techniques, and in particular electro-Fenton (EF) over Fenton. The crucial point is that with EF, hydrogen peroxide (required to initiate the Fenton reaction) is no longer needed, since it is generated by reduction of dissolved oxygen in a two-electron mechanism favored on carbon-based electrodes. This is one of the reasons why, in this study, we used a commercial carbon felt as the cathode.

These points have been clarified in lines 62 to 66 and in lines 80 to 81 of the document, as follows:

Lines 62-66: “The main advantage of electrochemical techniques over chemical techniques, and particularly EF over Fenton oxidation, is that with EF, hydrogen peroxide (H₂O₂) (required to initiate the Fenton reaction) is no longer needed because it is generated by reduction of dissolved oxygen in a two-electron mechanism on carbon-based electrodes”

Lines 80-81: “This is one of the reasons that led to using a commercial carbon felt as the cathode in this study”.

Comment 3: The concentration of the RhB should be optimized.

Reply 3: In this type of study, it is indeed important to work on a target molecule concentration that is consistent with both the environmental reality and the problem of the study.

To investigate the influence of the initial RhB concentration, two different RhB concentrations have been studied: 0.01 mM (4.79 mg L-1) and 0.1 mM (47.9 mg L-1). These concentrations are in the same range of the EC50 and LD50 found in the literature, respectively [9] and this has been already mentioned in lines 179-180. Moreover, one of our objectives was to determine the reaction pathways of dye degradation. For these experiments, we selected a Rhodamine B concentration of 0.1 mM which allows to quantify, with a sufficient accuracy, the concentration changes as a function of time treatment. Actually, at too low concentrations, the fast kinetics does not allow to identify all the intermediates.

For all these reasons, we consider that our both selected RhB concentrations are pertinent, with regard to the work objectives.

Comment 4: English language and grammar of the manuscript are recommended to be improved.

Reply 4: English language and grammar have been corrected by a native speaker.

Comment 5:  The format of the references should be checked and revised according to the author’s guideline of the journal.

Reply 5:  The format of the references has been checked.

Reviewer 3 Report

Comments and Suggestions for Authors

This work put in front a valuable and complete study about dye degradation. Unlike most similar studies dealing with RhB degradation, this one indicates the mechanistic pathway involved and evaluates the toxicity of the intermediates. This work can be published after minor revision taking into account the following:

-Please, add some lines about the practical application of this study

Author Response

Comment: Please, add some lines about the practical application of this study.

Reply: 

The practical applications of this study are focused on the treatment of effluents from the textile industry, which is a major user of dyes, particularly in developing countries for which the problem of treating effluents at source is paramount in view of the possible absence of sewerage networks. But these electrochemical technologies are also applicable to the treatment of domestic effluents for tertiary treatment, after a membrane bioreactor as we have already published [new 41].

These points have been clarified in line 268 to 277 of the document, as follows: “The complexity of real wastewater, with its variable organic load and diverse pollutants, will undoubtedly influence the degradation performance. Previous studies confirmed the effectiveness of these processes for treating complex matrices, underlining their potential for practical applications [new 40, new 41]. These findings are particularly relevant for the textile industry, a major cause of dye-contaminated effluents, particularly in developing countries where the problem of treating effluents at the source is crucial due to the possible absence of sewerage systems. Furthermore, these electrochemical technologies are also applicable to the tertiary treatment of domestic wastewater following membrane bioreactor treatment [new 42]”.

Round 2

Reviewer 1 Report

Comments and Suggestions for Authors

The manuscript can be accepted.

Reviewer 2 Report

Comments and Suggestions for Authors

The manuscript is recommended to be accecpted.